# Conformity and tradition are more important than environmental values in constraining resource overharvest

Glenn Wright[1]*, Carl Salk[2,3], Piotr Magnuszewski[4,5], Joanna Stefanska[4], Krister Andersson[6], Jean Paul Benavides[7], Robin Chazdon[8,9]

1 Department of Social Science, University of Alaska Southeast, Juneau, Alaska, United States of America, 2 Southern Swedish Forest Research Centre, Swedish University of Agricultural Sciences, Alnarp, Sweden, 3 Institute for Globally Distributed Research and Education, Gothenburg, Sweden, 4 Centre for Systems Solutions, Wrocław, Poland, 5 International Institute for Applied Systems Analysis, Laxenburg, Austria, 6 Institute of Behavioral Science, University of Colorado, Boulder, Colorado, United States of America, 7 Instituto de Investigaciones Socio-Económicas, Universidad Católica de Bolivia, La Paz, Bolivia, 8 Department of Biology, University of Connecticut, Storrs, Connecticut, United States of America, 9 Tropical Forests and People Research Centre, University of the Sunshine Coast, Sippy Downs, Queensland, Australia

* gdwright@alaska.edu

**Data Availability Statement:** Data are available from the Inter-University Consortium for Political and Social Research (ICPSR): Wright, Glenn.

## Abstract

We present the results of a hybrid research design that borrows from both experimental techniques—experimental games—and observational techniques—surveys—to examine the relationships between basic human values and exposure to natural ecosystems, on the one hand, and collective action for resource governance, on the other. We initially hypothesize that more frequent exposure to forests, and more pro-environmental values will be associated with more conservation action. However, we find that other values—tradition and conformity—are more important than pro-environmental values or exposure to nature. Our results imply that resource governance is likely to be more successful where resource users hold values that facilitate cooperation, not necessarily strong pro-environmental values.

## Introduction

Although contemporary scholars have constructed a substantial literature examining the causes of effective resource governance [1–5] a great deal of extant research tends to assume that resource users are rational egoists, seeking to maximize individual utility [6]. To the degree that scholars have examined the values that motivate successful, sustainable resource use, they have tended to assume that the most important motivations have been pro-environmental values; people work to conserve resources because they value those resources implicitly [7–9]. Scholars have also suggested that values should be incorporated into technical resource governance decisions through processes like ecosystem service valuation [10–12]. However, individual or shared values may have a pervasive impact on policy even in the absence of technical processes like cost-benefit analysis or ecosystem service valuation. Values may be an

Conformity and tradition are more important than environmental values in constraining resource overharvest. Ann Arbor, MI: Inter-university Consortium for Political and Social Research [distributor], 2022-07-28. https://doi.org/10.3886/E176461V1.

**Funding:** This research was funded through US National Science Foundation Coupled Natural and Human Systems Grant 1114984, "CNH: The Emergence of Adaptive Governance Arrangements for Tropical Forest. Ecosystems." The funders had no role in study design, data collection and analysis, decision to publish, or preparation of the manuscript.

**Competing interests:** The authors have declared that no competing interests exists.

important driver of resource governance and may have broadly felt impacts on most policy decisions, but these impacts have rarely been studied, despite a large literature in social psychology examining the role of values in motivating behavior [13].

Here, we present the results of a hybrid research project combining aspects of observational and experimental data-gathering which allows us to evaluate the impact of personal values on resource governance. We examine several types of values and motivations, including exposure to nature, pro-environmental values and pro-social values (such as valuing tradition and conformity) to see if these factors drive resource governance decisions. Our results suggest that pro-social values and not pro-environmental values or other factors are most likely to encourage effective natural resource governance by facilitating effective cooperation.

## Values, environment, and resource management

A great deal of research has been based on the theory of personal values developed by Shalom Schwartz and collaborators [14, 15] or the 'Big Five' conceptualization of personality traits [16], two closely-related schema [17]. Other similar frameworks for measuring values also exist, such as "Moral Foundations," which measure moral values, norms, identities and other characteristics [18, 19]. This research has generally focused on exploring how altruistic, pro-environmental values may drive environmental attitudes and behaviors. Personal values are usually defined as concepts or beliefs which drive goals or behaviors, transcend specific situations, and guide behavior and evaluation of events [20]. Values are more persistent and more deeply held than norms, and apply more broadly than scripts, but are typically not formalized as rules of behavior. Scripts are normally defined by social psychologists as sequences of actions expected in a particular situation involving interaction between individuals. Values may be held by single individuals, as "personal values," [21] and may be widely held in society, in which case they are sometimes called "cultural values", "shared values," or "social values" [22–24]. The relationship between individual, and collective (shared, plural, social, cultural) values is contested, with some scholars arguing that values of all types are deeply held and slow-changing [15] and are closely related to individually-held personal values [25] while others argue that shared, social, or cultural values exist separately from personally-held values and may only emerge through discursive processes [26, 27].

In social psychology, scholars predict that pro-environmental values—sometimes associated with "universalistic altruism," an altruistic feeling towards everything outside the individual–may predict pro-environmental attitudes [28, 29] support for the environmental movement [30], and pro-environmental behaviors [31] by promoting pro-environmental norms and beliefs.

Psychologists also argue that personal values may drive a wide range of other human behaviors, including ones affecting natural resource governance. Cultural values have proven meaningful in explaining national cultural differences, even between countries with similar language, religion and history [25, 32]. Values underlie political preferences in some countries [33], though not all [21]. Values are also important in many other realms, including risk perception [34], workplace managerial behaviors [35], religiosity [36], "moral" characteristics [37], and gender equality [38].

Much work on values in environmental governance examines the ways that values at the individual or group level might be elicited or developed in order to better inform technical decision-making processes like cost-benefit analysis or ecosystem service valuation [11, 12]. These scholars argue that decision-makers will tend to make better decisions when working to manage resources if their decisions take into account the values of resource users and other stakeholders [39, 40].

However, researchers have noted a growing gap between the policy-making processes and outcomes imagined in work on ecosystem services valuation and policy outcomes in the real-world [41]. Partly, this may be because policy makers in many settings do not use cost-benefit analysis or ecosystem service valuation techniques; many policy making processes are democratic or discursive and driven by local institutions [7, 42]. And policy-makers—themselves often resource users or stakeholders, not neutral brokers—likely hold values that drive decisions about resource use or harvest or conservation.

If values are related to such diverse personal choices as political preferences, religiosity and gender equality [33, 36, 38], perhaps they are also linked with collective action; might they help (or harm) people's abilities to act collectively, for example, to create norms and institutions for resource governance? Such institutions require a certain degree of self-sacrifice, self-constraint, and cooperative behavior on behalf of resource users [3, 43]. If pro-environmental individuals are more likely to practice self-restraint when interacting with natural resources [31, 44], does this mean that those individuals are also more willing to cooperate in social dilemmas? Are pro-environmental resource users more willing to contribute to the creation of institutions, even when these infringe on the users' personal freedoms and interests?

To address these questions, we investigate five of the ten personal value complexes proposed by Schwartz [15] for which links to individuals' resource governance behavior can be expected: (1) power, (2) self-direction, (3) universalism and (4/5) conformity/tradition. Two of these values are associated with self-interest; The "power" value is about the importance of social status, prestige, control or dominance over people or resources. In a resource-governance situation, power-valuing individuals may harvest heavily to increase their income, gaining status or control over resources or other people. Individuals valuing "self-direction" place importance on independent thought and action, and view exploration, creativity, freedom, and independence favorably. These are self-regarding values which could plausibly promote greater cooperation by encouraging individuals to promote their own enlightened self-interest by encouraging rules and norms for cooperation. However, self-directed individuals may also be less likely to cooperate with others because—by definition—hey place personal freedom above cooperation. The "universalism" value is about the importance of understanding, appreciating, tolerating and protecting the welfare of all people and of nature. This value complex is the most closely related to Agrawal's "environmentality" and has been most frequently associated with pro-environmental attitudes and behaviors [7]. We hypothesize that more universalistic individuals will harvest less because of their greater concern for others and for the environment. Finally, "conformity" and "tradition" are two closely-related values. Conformity is held by individuals who value adherence to social norms oppose actions which will tend to violate such norms. "Tradition" is about the respect, commitment and acceptance of customs and ideas held by one's culture and/or religion. We hypothesize that individuals who value conformity and tradition will be more likely to value group agreement and the development of shared norms, making them more likely to behave cooperatively with others, diverge less from group harvesting norms and harvest less than nonconformist individuals.

Beyond values, we are also interested in how exposure to natural ecosystems affects sustainable resource governance decision making. Spending time in local ecosystems leads to greater knowledge of their ecological features, which is hypothesized to drive sustainable resource use. Scholars have presented evidence that exposure to nature may lead to pro-environmental values or behaviors [45, 46]. However, studies linking nature exposure with values are to date rare, providing a limited but tantalizing glimpse into the subject. Frost [47] found that middle and high school aged students in Florida and Peru who spend more time in forests have a greater sense of fascination and wonder in nature, and less aversion or fear of forests. Depending on a resource for outdoor recreation (e.g. rivers for kayaking) leads to greater awareness of

environmental issues surrounding that resource, even when they do not directly impact recreational opportunities [46]. However, countervailing patterns are also possible, particularly when one considers why people are spending time in nature. A survey of Austrian forest owners found that those who live in cities spent less time in forests than rural owners, but their appreciation of forests was more likely to be naturalistic than economic [48]. We aim to further flesh out this body of scholarship by testing whether study participants who spend more time in real-world forests harvest less in the game microcosm.

In this study, we ask how values and forest experience affect resource-use behavior using a hybrid research methodology combining observational and experimental techniques in a game-like behavioral simulation at eight sites in Bolivia and Uganda. We describe the behavioral simulation in detail below, under "Methods." We find significant relationships between tradition, conformity, and resource extraction choices. This suggests that values can play an important role in the development of institutions for resource governance. At the same time, we find evidence that pro-social values facilitating rule following are much more important than pro-environmental values or direct experience with forests in driving resource outcomes.

## Methods

We adopt a hybrid approach combining three interrelated methods. The first is a series of survey questions measuring cultural values developed by Schwartz and colleagues [49]. The second is a natural resource management game. The third is a series of survey questions about experience with real-world forests, and individual characteristics including demographics. The details of these tools are described in following the description of our study sites below. Our hybrid approach is not strictly experimental in the sense that we do not randomly treat subjects with some stimulus. However, the behavioral simulation activity allows us to examine the effects of subjects' values and experiences on real behaviors while ruling out many possible sources of endogeneity. By measuring individual-level characteristics, such as values, we can test relationships between a number of these characteristics and individual-level behaviors much more efficiently than using a series of experiments (for example, running four separate experiments to test the effect of each value complex, rather than our single simulation). While our approach may not allow us determine causality in all cases, it can generate helpful results which eliminate many sources of potential endogeneity and may guide future analysis and experimental research.

All participants provided informed consent verbally before participating in our simulation activity or surveys. Institutional Review was provided by the University of Colorado Institutional Review Board.

Games and simulations similar to the one we use here are a useful technique for investigating human interactions. Though by necessity they place participants in a stylized, simplification of some real-world situation, these simplifications can be structured such that they involve real human interaction and involve real-world consequences. For example, the simulation we construct includes real monetary payoffs for participants that vary depending on their individual and collective choices. And though stylized, the simulation places participants in a real collective action problem (similar to a prisoners' dilemma) which resembles real-world resource management dilemmas in important ways. We describe these characteristics below (in "The Forest Management Game." Experiments, simulations, and games have been used widely in the social sciences to study human interactions in a range of settings and around a range of questions. For example, Henrich et al. [50] use three simple games ("Dictator Game," "Ultimatum Game," and "Third-Party Punishment Game") to illustrate differences in cooperation across communities of differing sizes and levels of market integration. Janssen, Anderies

and Cardenas [51] use experiments structured as games in both the laboratory and field settings in the developing world to study how agriculturalists can sustain cooperation in settings with collective irrigation systems, and Andersson et al. [52] used game-like experiments to examine the effects of payments for ecosystem services on forest conservation behaviors. The game we use here is based on experiments carried out by Cardenas, Janssen and Bousquet [53] who sought to examine the role different types of rules play in successful resource governance. Many similar studies have often been used to demonstrate and measure the ways human behaviors diverge from neoclassical economic theory [54], for example by demonstrating that individuals are much more likely to cooperate when they are allowed to communicate. Levitt and List [55], List [56] and Anderies et al. [54] provide excellent descriptions of the broad literatures in Economics, Political Science, and Psychology and Sociology that use experimental games and simulations to illustrate surprising characteristics of human interaction. Games, behavioral simulations and experiments are not a substitute for other forms of empirical data gathering, but are useful complements to qualitative interview techniques, surveys, participant observation and other empirical strategies.

## Study regions

To examine the effects of social values on resource governance behavior, we chose eight forest-dependent sites, four in Uganda and four in Bolivia, intentionally selected for their ethnic and historical differences, with the goal of maximizing variation in personal and cultural values. Each site is a forest-dependent community located in a rural area. We used the International Forestry Resources and Institutions (IFRI) database of forest-dependent communities to help select field sites in tropical forest settings that met the above criteria. Our site selection process proved successful, as our participants displayed a range of different values, although values were more consistent than expected within each of the two countries, something we discuss further in the results and discussion sections. Due to national demographic differences, the Bolivian sites are consistently more forest-rich and more sparsely populated than the Ugandan sites, while the more densely populated Ugandan sites tend to experience more forest degradation and deforestation. These eight sites are mapped in Fig 1.

The four Bolivian sites are all in lowland areas with humid tropical forests. San Isidro, our first site, is a small, well-organized village near Rurrenabaque, a regional center, in the lowlands about 250 km north of La Paz (Bolivia's largest city). Simay is a larger community near the town of Palos Blancos in the lower Yungas is 150 km northeast of La Paz. TIM Ivirgarzama is close to a fast-growing town about 140 km northeast of Cochabamba. Cururu is a very remote indigenous village in northeastern Bolivia. San Isidro and Simay are ethnically diverse, while Ivirgarzama and Cururu are more homogenous, although the surroundings of Ivirgarzama are relatively more diverse.

All of the selected Ugandan sites depend heavily on increasingly-threatened forest resources. Mpanga is an agricultural community about 50 km west of Kampala (the largest city and national capital) on a major highway. The site includes two villages and is known for making wooden drums, originally for cultural ceremonies but now mostly for sale to passing tourists [57]. Nyabyeya is a more remote agricultural village near Lake Albert suffering from deforestation and forest degradation. This local deforestation stems largely from illegal charcoal production and clearing for agriculture. Sango Bay is a site located in the south, near the Tanzanian border and Lake Victoria. It experienced depopulation during the AIDS epidemic and Uganda's civil wars and remains more sparsely populated than most areas of southern Uganda today. Nearby forests are extensive and relatively healthy and the community has good relations with Uganda's National Forestry Agency. Finally, Echuya is a village located in

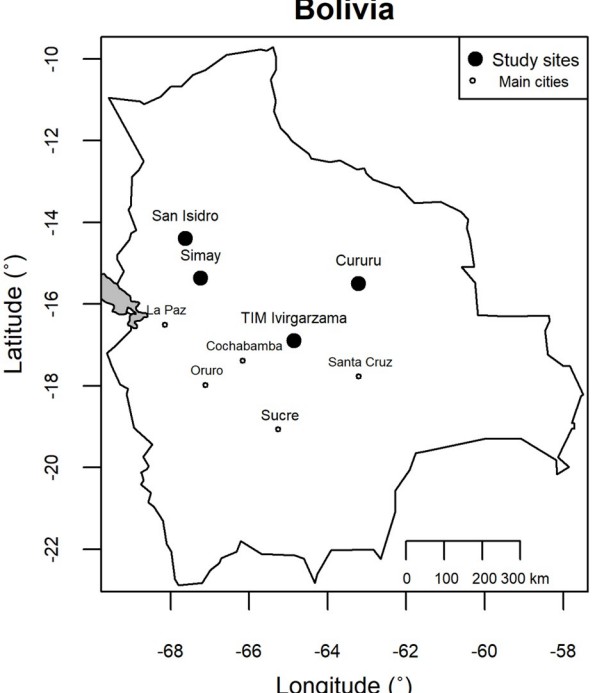
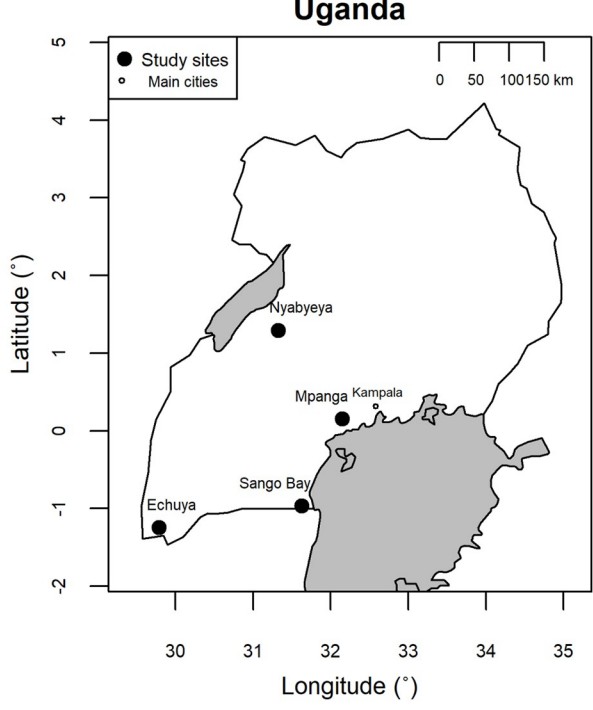

**Fig 1. Location of study sites.**

far southwest Uganda, near the border with Rwanda. Echuya is ethnically diverse, including a substantial contingent of Batwa people who are traditionally forest dwellers, but who were evicted from their nearby lands in the 1980s for the creation of a national park. This site is closer to major cities in Rwanda and the Democratic Republic of Congo than to Kampala, but is on an important highway. It also has the least organized local governance institutions of our Ugandan sites.

## Personal value survey

To measure personal values, we adopt Schwartz's theory [49] and its accompanying battery of survey questions. Although we are unaware of any application of this technique in Bolivia or Uganda, it has now been tested and confirmed to provide similar results in dozens of countries [13]. For tractability in the field, we used the short version of this survey, which has 20 brief descriptions of a person's value-oriented traits. The respondent must rate their own similarity with each statement on a 6-point Likert scale. After field work, these responses were used to calculate each participant's score on 10 different value scales, following Schwartz's methodology [49]. This survey battery was carried out once simulations/games had been completed.

## Pre- and post-game surveys

In addition to the survey on values, pre- and post-game surveys covered several topics, including questions about how much time one spends in real forests, what people do in the forest, and demographic information used as control variables in the analyses described in the statistical methods. A substantial body of research has found that time spent in real-world forests is a useful proxy for a wide array of conservation-related attitudes, including recreational activities as well as consumptive activities like timber harvest, though most research in this area has

focused on recreational activity in the United States. Our survey asked two separate questions about time spent visiting forests. The first was the number of times the respondent had visited the forest in the last month. The second was how much time they spent on an average forest visit, on a five-point ordinal scale. To estimate each participant's total time in the forest in the last month, we multiplied the number of visits variable by the midpoint of their average visit length category (in hours). Because the distribution of this last variable is skewed, we transformed it as $log_{10}(monthly\ hours\ in\ forest + 1)$.

### The forest management game

Our behavioral simulation is a dynamic common-pool resource management game modeled after the "forestry game" developed by Cárdenas, Janssen and Bousquet [53]. In each session, local contacts helped us assemble a group of eight players. These players manage a resource—a stylized forest—over 15 "rounds." Each round, individual players choose how many "patches" to harvest, and whether to monitor the harvest of other players or "sanction" (punish) other players. Unlike in typical experimental economics games, the resource is dynamic, so the forest is not the same size in every round. Rather, the first round of the game always begins with the forest in the same state (100 of 100 possible patches forested), but changes depending on the players' cumulative harvesting. Harvesting reduces the size of the forest, but the forest also regenerates, growing back at a rate proportional to the number of unharvested forested "patches." Consequently, the Cárdenas, Janssen and Bousquet game captures the ecological dynamics of real-world resources better than behavioral experiments with static resources, although this also complicates our analyses, a point discussed in the statistical methods. The game has two key characteristics. First, the game places players in a "prisoners' dilemma"-like situation with a strong individual incentive to overharvest, but where collective earnings are highest if everyone restrains their harvest. Second, consequences of decisions in one round affect future rounds (as is the case with most shared resources in the real world).

In each round, participants individually decide how much to harvest (up to a limit related to the size of the forest) though individuals are free to communicate and groups may (and often do) attempt to coordinate their harvest decisions. Individual harvesting decisions are private, not visible to other participants, but once participants' harvest decisions are made, they are able to observe the group's total harvest and new forest size following regeneration in each round. Half of the game groups included a Payment for Ecosystem Services (PES) treatment; although it is not relevant to the questions asked by this study, we do control for it in the data analysis stage by including a control variable in our statistical models (see statistical methods). Harvesting "patches" earns players money, which are converted to local currency at the end of the game at a rate based on local income levels. Players can also pay some of their earnings to monitor other players' harvesting behavior or to "sanction" (punish) other players. Two observers observe players' actions during the game and assist participants if needed.

Although optimal strategies in this game are slightly complicated by the dynamic resource, participants readily grasped basic strategy tradeoffs. The game's Nash equilibrium is quite simple; individual gain is always maximized by harvesting the maximum allowed. The Pareto optimum (which we refer to as the "social optimum") is for everyone to harvest two trees/round, except in the last two rounds of the non-PES treatment, and leads to much greater overall payoffs than Nash-strategy harvesting. However, if players overharvest and the forest shrinks substantially, determining the social optimum becomes more complicated. Under most circumstances, players can maximize their total harvest (as a group) and therefore approach the social optimum by temporarily restraining harvesting until the forest regrows to maximum size. A full game manual is found in the S1 File.

## Statistical methods

We use multivariate ordinary least squares regression to evaluate links between personal traits and harvesting behavior in our games, both with and without control variables summarized in Table 1. We present multiple models for two reasons. The first is the complexity of evaluating harvesting in a game with a dynamic resource. Because harvesting limits depend on the state of the resource, we use both absolute harvest (Model 1) and harvest as a fraction of the maximum allowable amount (Model 2). Neither of these outcome variables is ideal under all circumstances; a smaller forest necessarily limits harvesting choices and also makes the same harvest value a bigger percentage of the allowable harvest. The second reason for multiple models is that our hypothesized relationships between different value scales and harvesting require different outcome variables to test. In particular, we expect self-directed individuals to more frequently deviate from group mean decisions. For this reason, in Model 3 our response variable is the absolute value of the difference between an individual's harvest and the group average in a round, averaged across rounds. This response variable can be seen as a measure of willingness to go along with collective action. Finally, in Model 4 we use deviation from the social optimum as the dependent variable (typically two patches per round, but see the analysis of optimal strategies in the game description). Deviation from the social optimum is a measure of effective collective action. In particular, it requires two steps to achieve a low value. A group must determine the social optimum, and also adhere to it collectively—i.e., a group must determine the optimal level of harvest, agree to collectively harvest at that level, and carry out their agreement not to individually over-harvest. Groups with more difficulty in acting collectively will tend to deviate more from the social optimum. In general, then, this measure is conceptually distinct from the level of harvest, but groups which harvest nearer the social optimum will also generally tend to have lower harvest, controlling for forest size or condition (and thus maximum allowable harvest).

The personal trait values included in the models include our five Schwartz values of interest (power, conformity, tradition, universalism, and self-direction) and the natural logarithm of self-reported forest visitation (computation of variable described above). As control variables, we use the average condition of the "game forest" across all rounds, and a dummy variable for

**Table 1.  Variable summary statistics.**

| Variable | N | Mean | Std. Dev | Min | Max |
|---|---|---|---|---|---|
| Harvest | 128 | 1.544 | 0.618 | 0.000 | 3.800 |
| Power | 128 | 0.767 | 1.126 | -1.619 | 4.286 |
| Conformity | 128 | -0.197 | 0.873 | -2.238 | 2.857 |
| Tradition | 128 | -0.268 | 0.770 | -2.238 | 1.714 |
| Universalism | 128 | -0.415 | 0.572 | -1.905 | 1.333 |
| Self-direction | 128 | 0.373 | 1.037 | -1.524 | 3.381 |
| $Log_{10}$(hours/month in forest) | 113 | .745 | .528 | 0.000 | 1.785 |
| Average forest size | 128 | 57.338 | 26.102 | 20.600 | 100.000 |
| Country (Bolivia = 1, Uganda = 0) | 128 | 0.500 | 0.502 | 0.000 | 1.000 |
| Treatment (PES = 1, no PES = 0) | 128 | 0.500 | 0.502 | 0.000 | 1.000 |
| Harvest: % of Max. Harvest | 128 | 0.360 | 0.177 | 0.000 | 0.910 |
| Harvest: Absolute Deviation From Group Average | 128 | 1.085 | 1.215 | 0.000 | 6.047 |
| Harvest: Deviation from Social Optimum | 128 | 0.092 | 0.097 | -0.091 | 0.428 |

Summary statistics of variables used in our models. The lower sample size for the hours/month in forest variable was due to occasional non-response in the post-game survey.

country. We also include a dummy variable to differentiate PES from non-PES games (coded 0 for non-PES and 1 for PES), though that experimental treatment is not the subject of this analysis and generally has no statistically significant impact in the results we present here.

We used a series of post-estimation checks to ensure the robustness of the results reported here. In general, residuals were normally distributed and lack signs of heteroskedasticity or non-linearity when plotted against fitted values or independent variables, with the exception of model three—where mean absolute deviation from average group harvest is the dependent variable. There, some heteroskedasticity is present, so we use heteroskedasticity-robust standard errors. These models are also robust to the exclusion of high-leverage cases and outliers. Finally, these results are also robust to inclusion and exclusion of various combinations of demographic control variables including several different measures of wealth and gender, and country- and community-level dummy variables. We also test for multicollinearity in our models; we find some evidence of multicollinearity between the tradition and conformity values complexes, but when we remove one of these variables from the model, coefficient direction does not change and the remaining variable becomes more highly statistically significant, suggesting that any problems related to multicollinearity are leading to an under-not over-estimation of the magnitude of the relationships we find here.

Due to distributions that diverge from normality, differences in forest visitation patterns between countries and between genders were analyzed using Mann-Whitney-Wilcoxon tests.

## Values and forest visitation in our study populations

Although we attempted to increase variation in personal values by choosing field sites with different histories and ethnicities, we found that values tended to cluster by country (Fig 2). Values are fairly consistent within countries and sites, but quite different between the two countries in our study. Bolivians tend to value power and conformity most highly. In Uganda, values are much more individualistic and self-directed (Fig 2), with much more variation in harvesting behavior within groups. In general, the consistency of measured values within each country appears to further validate this approach. However, there is still enough variation among individuals for statistical analysis of our results.

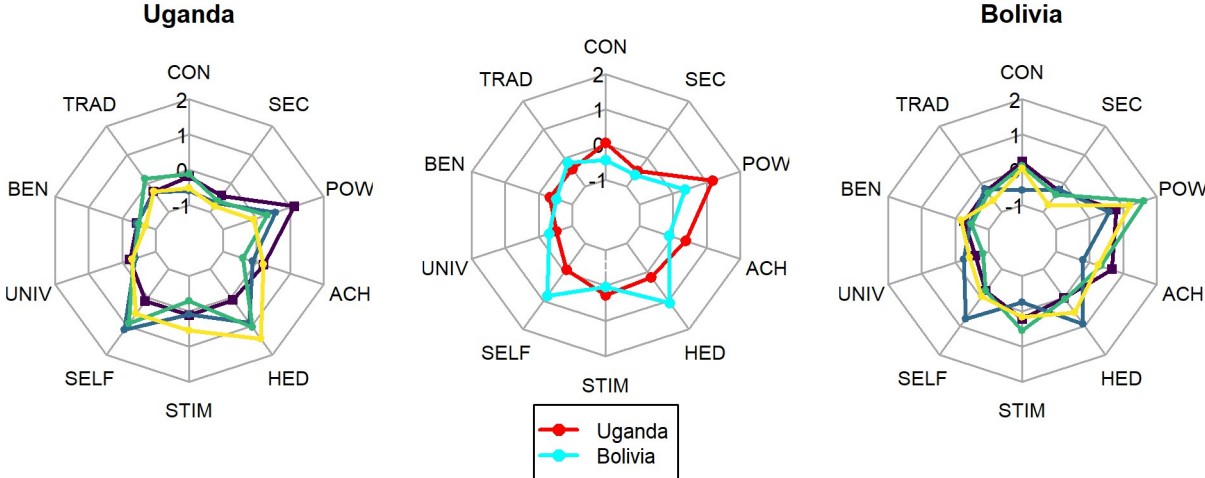

**Fig 2. Different value complexes in Bolivian and Ugandan sites.** The outer panels show the mean values across all 16 participants in each of the four sites (different colored lines) in Uganda and Bolivia. The center panel shows averages across all four sites in each country. Key to value complex names: CONformity, SECurity, POWer, ACHievement, HEDonism, STIMulation, SELF-direction, UNIVersalism, BENevolence, TRADition.

On average, Bolivian participants visited forests much less frequently than Ugandans (1.8 vs. 4.7 visits/month; $p = .0009$, W = 1219, but the individual visits made by Bolivians were significantly longer, 7.7 vs 1.7 hours; $p < .0001$, W = 2773.5. This and all subsequent statistics in this paragraph are from Wilcoxon tests). Taken together, Bolivians spent on average nearly twice as much time in the forest as Ugandans (15.9 vs. 8.9 hours/month), although these values were not statistically distinguishable ($p = .8896$, W = 1682.5). In Bolivia, men spent slightly, but non-significantly, more time in the forest than women (17.1 vs. 14.9 hours/month; $p = .1998$, W = 547.5), whereas Ugandan participants showed the opposite pattern. Gender differences in Uganda were also statistically insignificant at $p = .05$ (5.5 vs. 11.8 hours/month; $p = .0824$, W = 284.5). In these comparisons of total time in forest, very similar results were found for t-tests on the log-transformed forest visitation variable.

## Statistical analysis of personal trait impacts on harvesting

Contrary to our expectations, the power and self-direction value complexes showed no statistical association with resource harvesting behavior (Table 2) in any of our models. The self-direction complex was only associated with harvesting behavior in model 2 where it showed the opposite of expected pattern; more self-directed individuals harvested less (Table 2). However, as predicted, conformity and tradition were both associated with lower harvest in several models (Table 2). Our strongest result was on the Tradition complex—tradition was negatively associated with the dependent variables in models 1, 2 and 3 ($p < .05$), while conformity and universalism were significantly ($p < .05$) associated with lower harvest in only the first model, where the dependent variable is unadjusted average harvest. Substantively, holding other variables constant, an individual near our lowest observed value on Conformity (about -2) would

**Table 2. Regression models.**

| | Model 1 | Model 2 | Model 3 | Model 4 |
|---|---|---|---|---|
| | Harvest | Harvest: % of Max. Harvest | Harvest: Absolute Deviation from Group Average | Harvest: Deviation from Social Optimum |
| Power | -0.04 (0.06) | -0.01 (0.02) | 0.10 (0.09) | -0.00 (0.01) |
| Conformity | -0.14 (0.07)* | -0.03 (0.02) | 0.00 (0.10) | -0.01 (0.01) |
| Tradition | -0.22 (0.08)** | -0.05 (0.02)* | -0.27 (0.12)* | -0.02 (0.01) |
| Universalism | 0.23 (0.10)* | 0.06 (0.03) | 0.27 (0.16) | 0.03 (0.02) |
| Self-direction | -0.08 (0.06) | -0.03 (0.02) | 0.02 (0.09) | -0.01 (0.01) |
| Forest visitation | 0.08 (0.12) | 0.08 (0.03)* | 0.17 (0.17) | 0.02 (0.02) |
| Bolivia | 0.17 (0.14) | 0.02 (0.04) | -1.77 (0.22)*** | -0.05 (0.02)* |
| Forest condition | $-3.97 \times 10^{-3}$ ($2.26 \times 10^{-3}$) | | $-3.55 \times 10^{-4}$ ($3.43 \times 10^{-3}$) | $-2.50 \times 10^{-3}$ ($3.59 \times 10^{-4}$)*** |
| Treatment (PES) | -0.10 (0.12) | -0.07 (0.03)* | 0.33 (0.17) | -0.03 (0.02) |
| Constant | 1.28 (0.20)*** | 0.34 (0.04)*** | 1.67 (0.31)*** | 0.34 (0.03)*** |
| $R^2$ | 0.19 | 0.18 | 0.55 | 0.49 |
| $N$ | 113 | 113 | 113 | 113 |

The first six independent variables are individual traits; 'Forest visitation' is the log of how many hours the respondent spent in the forest in the last month. Values in parentheses are standard errors. Note that the forest condition variable is left out of model 2 because it is used in the calculation of the response variable. Significance codes:

* $p < .05$;

** $p < .01$

*** $p < .001$

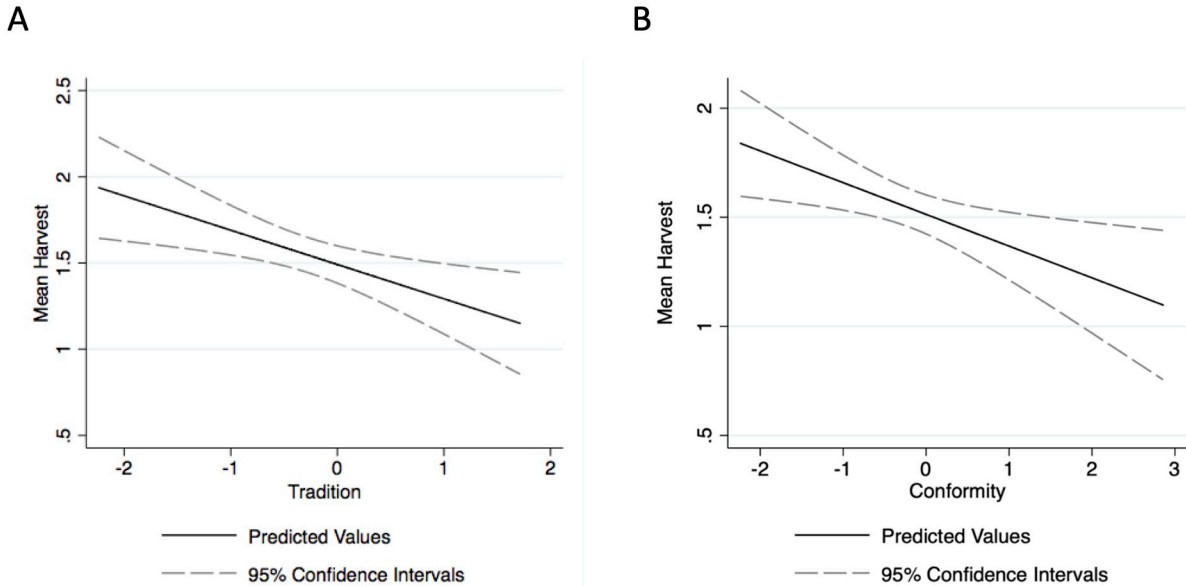

**Fig 3. Predicted relationship between (A) Tradition and (B) Conformity and Harvest in Model 1 (see Table 2).** The dashed lines show 95% confidence intervals of this relationship.

harvest about one more patch per round compared to an individual near our highest observed value (about 3, see Fig 3). Likewise, an individual near the lowest observed level on Tradition (near -2) would harvest about .75 more patches per round than an individual valuing Tradition near our highest observed value (about 2, see Fig 3).

We also wondered if the relationship between values and harvesting would be stronger if we limited our analysis to the first few rounds of our simulations; it is possible that in later stages of the game, players will have developed institutions that will impact harvesting more than values. Therefore, we re-ran our statistical models several times, including, respectively, rounds 1 through 3, 1 through 4 and 1 through 5. In general, results of these models are similar to those reported above, though as we reduced our number of observations by excluding rounds, levels of statistical significance tended to decrease.

We were surprised to find that time spent in forest had no discernible relationship to in-game harvesting behavior in any of the models (Table 2).

## Discussion

In our first model, we do find that Universalism is associated with harvesting behavior, but the relationship we find is the opposite of what we expected. Universalism is *positively* associated with harvest, so individuals with higher values on Universalism harvest more, on average, than individuals who value Universalism less highly. This is surprising, because Universalism is the complex of values that includes concern for the natural environment—we would expect that individuals placing greater value on the environment would harvest less. This may be a statistical fluke, a result of randomness or measurement error. A second possibility is that Universalistic individuals' regard for others around them leads them to more income-seeking behavior in an effort to earn money which they can use to benefit others. A final possibility is that harvesting behavior is dominated primarily by other values (tradition, conformity) leaving universalistic or pro-environmental values. To examine these latter possibilities, we re-ran our

statistical models using only the component of the Schwartz Personal Values Questionnaire asking about concern for the environment to see if players exhibiting high levels of pro-environmental values harvested less than other players. We found no statistically significant relationship between this measure of pro-environmental values and harvesting behavior. This result suggests that pro-environmental values may not be associated with harvesting because collective harvesting decisions are dominated by other factors (such as Conformity or Tradition values).

We conclude from these statistical results that Tradition and Conformity are the two values which are most likely to drive pro-environmental behavior, probably because individuals and groups with high levels of these values are more likely to develop and adhere to shared norms about resource governance. An alternate interpretation is that reliance on tradition may help resource users efficiently manage the complex cognitive demands of managing their relationships with ecosystems and their communities [41]. We view this result as especially interesting because it suggests that pro-social values are more important for effective natural resource governance than pro-environmental values, contrary to the argument presented by Agrawal [7]. It is also consistent with the literature on personal values which suggests that conformity, tradition, and stimulation are the three values most likely to drive differences in individual behavior [58]. That these results hold strongly through the game suggests that, to the extent institutions have an impact on harvesting behavior, institutions within the game may also be driven, at least in part, by more conformist or traditional values.

Our results show that the amount of contact with real forests has relatively little effect on the harvesting behavior of players during the natural resource simulation. This suggests that the particular reason for visiting forests may be more important than simply knowing how much time someone spends there, and time spent in nature for the purpose of resource extraction may be less likely to generate pro-environmental values or behaviors. At least as a first source of information, values may be much more effective as a predictor of how local resource users will react to incentives than simply knowing the extent of their forest contact.

Because of the research design of this project, it seems plausible to argue that personal values are causing the observed differences in players' behavior. Reverse causality is extremely unlikely because we measure players' values before they participate in our game: it would be difficult to argue that players' behavior in the course of the game is impacting their values, as assessed. Spurious correlation is a possibility, but we have been unable to develop a plausible-sounding explanation for a spurious relationship impacting both assessed personal values and harvesting. Consequently, we think we can plausibly argue that personal values are causing differences in observed behavior in our simulation game.

An additional concern might be that, by surveying our respondents prior to the simulation activity, we might be priming participants to respond in a particular way, and therefore, the relationships we find in our data may not reflect relationships in the real world. This is a real possibility that we considered when designing our research project, although we concluded that the disadvantages of possible remedies were outweighed by the logistical advantages of our approach.

A third question may be whether our results reflect any real-world relationships between values and collective action or values and environmental governance. Because participants were paid in cash for the choices made in our experiments, and because these games had the characteristics of a prisoners' dilemma, players found themselves in a very real collective action problem. At a minimum, then, our research likely measures the impact of values on collective action. The connection to environmental outcomes may be a bit less clear; extant research suggests that experimental research can closely resemble real-world behavior, but experimental behavior—and the ability of an experiment to replicate the real world—can be very sensitive to

relatively small decisions in experimental design [54, 55, 59]; it is possible that our stylized game "forest" was not sufficiently like a real forest to trigger participants' Universalistic values, for example, even though Universalistic values might drive behavior in a real forest. Once source of data on this question comes from our post-game survey. The evidence that we have at our disposal suggests that participants saw the game "forest" as much like a real forest—in our post-game survey, we asked participants whether the game forest resembled real forests that they have experience with. Most responses suggested that participants saw the game as very similar to real forests in many ways, often specifically citing dynamics related to over-harvesting and the ways in which other players saw the game forest primarily as a source of income (i.e., implying that many see real forests primarily as an economic resource rather than something to be valued in its own sake). Though this evidence can by no means be considered conclusive, it does suggest further that pro-social values are more important for effective resource governance than pro-environmental values.

The simulation approach we followed has important advantages but also a few disadvantages. As compared to traditional observational analysis, the behavioral simulation approach allows us to place each group of players in an identical setting. This helps us rule out many characteristics of the local setting—such as forest characteristics or other bio-geographical factors, or local, regional, or national institutional factor—as spurious drivers of resource governance outcomes. On the other hand, the simulation approach, like experimental research, raises certain questions about external validity; do participants behave in the simulation in the same way they would behave in the real world? However, this particular simulation places players in a *real* collective action problem with real-world consequences, and so allows us to draw clear conclusions about resource-use behaviors. As compared to conventional experimental techniques, our hybrid approach allows us to examine the relationships between individuals' observed, real-world characteristics and behaviors, and therefore may raise fewer concerns about external validity. But because we do not randomly treat our participants with some stimulus, some types of endogeneity cannot be definitively ruled out by our approach.

One interesting related question about our research design is whether we are measuring individual values and their possible impacts on collective action and resource governance or some sort of collective values complex (plural, social, cultural etc.). As noted above (in "Methods") we measured values at the individual level through surveys carried out with each simulation participant prior to the start of the simulation activity. Some scholars have argued that individual values are themselves a reflection of cultural and/or social context and that institutions may tend to empower those widely held social or cultural values which then impact individuals and the values that they hold individually [12]. On the other hand, shared values may only emerge through discourse and therefore, discursive structures are necessary for the development and elicitation of collective values [60]. Though our simulation design encourages discussion, values that may have been developed through discourse in the course of the simulation would not have been measured through the survey instrument we used to measure values, which preceded the simulation activity. On the other hand, we intentionally selected study sites to maximize the potential variation in aggregate (group, social, cultural, shared) values and observed consistency within sites (and within countries, see Fig 2). Unfortunately, our study design does not provide a great deal of leverage to examine the extent to which individual and shared values are linked.

On the other hand, our results do allow us to examine the ways in which values (whether individual or shared) drive collective action, a group outcome of individual decisions. Our simulation activity explicitly places participants in a prisoners' dilemma-like situation which, though stylized and somewhat abstract, shares essential characteristics with real-world resource management problems; notably, collective restraint stands to benefit the group

overall but individuals face strong incentives to over-harvest. The details of real-world timber harvest vary somewhat across our study sites, but in each case, community members routinely make decisions to harvest timber—for firewood, charcoal, or agricultural cultivation, for example—or to self-restrain in order to conserve forest resources for the long term. Our results suggest that the values we measured at the individual level (which may themselves be at least partly a result of shared values and value-laden institutions) have an impact on individual decisions and group-level cooperation for sustainable resource management. Thus, the values we observe at the individual level seem to drive decisions at both individual and group levels in our simulation and may therefore be associated with group-level outcomes in the real world.

## Conclusion

In this paper we describe the result of a study in which survey data on personal values is combined with a dynamic, strategic resource management simulation to help determine the effects of personal values on resource governance activities. Participants in eight field sites in Uganda and Bolivia were surveyed using an instrument developed to measure values. Then, participants took place in a natural resource governance simulation—a game—that allowed us to observe efforts to act collectively in a carefully controlled setting. This process allowed us to determine what values, if any, were associated with more effective collective action and natural resource governance. Initially, we expected that pro-environmental values would be most closely associated with increased cooperation and decreased harvest in our natural resource governance game. However, we found little evidence that pro-environmental values drive resource governance behavior. Instead, we found that two closely-related complexes of values —Tradition and Conformity—were negatively and significantly associated with harvesting behavior in the simulation environment.

These results are important for two reasons. First, this study is, to our knowledge, the first attempt to link personal values with collective action and resource management using quantitative techniques. Our results suggest that the emphasis some scholars have placed on values in resource governance is warranted, but perhaps has not focused on the most important values; quantitative analysis of values may be fruitful ground for future study. The specific emphasis that some scholars have placed on environmental values may be misplaced; environmental values may be much less important than values that encourage norm development and rule-following.

Second, our results may speak to the substantial literature on motivational crowding. Scholars have found that motivational "crowding in" and "crowding out" are not universal phenomena; in some settings and circumstances, applying external rules or incentives crowds out intrinsic motivation [61]. In other settings, similar rules and incentives do not lead to crowding out, and can even lead to a desirable motivational crowding in [62]. One explanation for these differences may be that similar types of incentives or rules affect resource users differently, depending on the different values those users hold. Some scholars have suggested that crowding in and crowding out effects may be a result of rules and incentives changing or priming the values resource users apply to a particular resource management problem. For example, a payment for ecosystem services might make resource users think of forests or fish in economic rather than spiritual or environmental terms [61]. If those resource users value economic outcomes—perhaps because they value hedonism or achievement—such a change might be desirable. If users do not value economic outcomes—perhaps because they value tradition or hold universalistic values—shifting the way users think about resource governance might lead to undesirable outcomes, such as unsustainable overharvest. Our results here do not speak directly to the effect of values on crowding in and crowding out, but they do provide

some evidence that values often motivate resource governance decisions. We therefore suggest that examining the relationship between values and crowding in and crowding out might be fertile ground for future research.

Scholarship has suggested that that taking local values into account when designing incentives and policy might lead to better outcomes [63]. Our research suggests that policy-making processes and the ability to sustainably manage resources is already substantially driven by values held at the individual or group level. For example, in settings where tradition and conformity are highly valued, institutions for rule making and rule following that are consistent with local traditions might lead to more consistent adherence to rules. Our results suggest, then—consistent with much extant research—that in many settings, local institutions developed by resource users may therefore be the best focus for natural resource governance. In other cases, where locally-developed institutions are absent or do not function well but where local resource users tend to value tradition and conformity, developing new institutions for resource-management collective action may be more successful compared to settings where users are less traditional and conformist.

## Supporting information

**S1 File.**
(DOCX)

**S1 Fig. Harvesting form.**
(TIF)

**S2 Fig. Monitoring form.**
(TIF)

**S3 Fig. Sanctioning form.**
(TIF)

**S4 Fig. Debt card.**
(TIF)

**S5 Fig. Cashier's sheet section 1.**
(TIF)

**S6 Fig. Cashier's sheet section 3.**
(TIF)

**S7 Fig. Cashier's sheet section 4.**
(TIF)

**S8 Fig. Decision box.** Picture of animal with participant's number.
(TIF)

**S9 Fig. Inside decision box.** Participant's number and pictures representing activities corresponding to given slot.
(TIF)

**S10 Fig. Income tokens.**
(TIF)

**S11 Fig. Observation protocol.**
(TIF)

## Author Contributions

**Conceptualization:** Glenn Wright, Carl Salk, Piotr Magnuszewski, Joanna Stefanska, Krister Andersson, Robin Chazdon.

**Data curation:** Glenn Wright, Carl Salk, Jean Paul Benavides.

**Formal analysis:** Glenn Wright, Carl Salk.

**Investigation:** Glenn Wright, Carl Salk, Krister Andersson, Jean Paul Benavides, Robin Chazdon.

**Methodology:** Glenn Wright, Robin Chazdon.

**Writing – original draft:** Glenn Wright, Carl Salk.

**Writing – review & editing:** Glenn Wright, Carl Salk, Krister Andersson.

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
