## [Decision Letter · Decision Letter 0]

1 May 2022

PONE-D-21-34547Conformity and Tradition are More Important than Environmental Values in Constraining Resource OverharvestPLOS ONE

Dear Dr. Wright,

Thank you for submitting your manuscript to PLOS ONE. After careful consideration, we feel that it has merit but does not fully meet PLOS ONE’s publication criteria as it currently stands. Therefore, we invite you to submit a revised version of the manuscript that addresses the points raised during the review process.

I do apologize for the delay in the editorial process. It has generally proven difficult to find reviewers and receive timely reviews for all papers and for that reason I have decided to proceed with a single review rather than delay any further. There are a number of key points in the review below that need to be addressed, in particular, regarding alternative approaches to the problem you are looking at based in factors such as cultural ecosystem services, etc.

We look forward to receiving your revised manuscript.

Kind regards,

Hisham Zerriffi

Academic Editor

PLOS ONE

Journal Requirements:

2. Please provide additional details regarding participant consent. In the Methods section, please ensure that you have specified (1) whether consent was informed and (2) what type you obtained (for instance, written or verbal). If your study included minors, state whether you obtained consent from parents or guardians. If the need for consent was waived by the ethics committee, please include this information.

Reviewers' comments:

Reviewer's Responses to Questions

**Comments to the Author**

1. Is the manuscript technically sound, and do the data support the conclusions?

Reviewer #1: Yes

2. Has the statistical analysis been performed appropriately and rigorously? 

Reviewer #1: Yes

3. Have the authors made all data underlying the findings in their manuscript fully available?

Reviewer #1: Yes

4. Is the manuscript presented in an intelligible fashion and written in standard English?

Reviewer #1: Yes

5. Review Comments to the Author

Reviewer #1: This is the first paper I’ve seen looking at both environmental values and gaming behavior so I appreciate the relative novelty of your methods. I, however, see your research design as having a western, homo economicus approach to studying people outside of the “weird” cultures in which much of the theories underpinning political science and economics developed. I’m referencing Heinrich’s definition of “weird”: western, educated, industrialized, rich, democratic. The focus on individual values and differences at the individual level seem less relevant that a focus on identifying shared, collective values in your study sites. In the end, your data points to pro-social behavior driving decisions more than individually-oriented motivations driving decisions. In your set up and discussion of results, I think you ought to engage more with literatures that look beyond the individual, such as cultural ecosystem services, relational values and social valuation of the environment, e.g.:

Kenter, J. O., et al. (2016). Shared values and deliberative valuation: Future directions. Ecosystem Services, 21, 358–371. https://doi.org/10.1016/j.ecoser.2016.10.006

Also, I encourage you to peruse:

Levine, J., Chan, K. M. A. & Satterfield, T. (2015). From rational actor to efficient complexity manager: Exorcising the ghost of Homo economicus with a unified synthesis of cognition research. Ecological Economics, 114, 22–32. https://doi.org/10.1016/j.ecolecon.2015.03.010

Another interpretation of your results, which the Levine et al 2015 paper prompted me to recall, could be that people are efficiency maximizers. You evidence shows that identifying or not with pro-environmental values doesn’t explain behavior that well your study. Relying on tradition and conformity reduces cognitive burden and likely helps people maintain good relations in their small communities.

A few details to address:

p. 3 last line: Scripts needs some introduction.

p. 6 Explain more in the intro about why a game microcosm sheds light on real world behavior. I now see that you reflect on this on p. 15 but I’d like to see some of this in the study’s set-up. I remain skeptical that how people behave in a game when they know they are being studied is a good proxy for real world behavior. A brief review of literature on how game behavior provides insight on real world behavior related to environmental management could convince me otherwise. It seems like quite a reductionist approach to understanding real-world forestry management and choices. It’s also not clear to me if deciding the amount of trees to harvest in your study sites is a collective or individual choice. My sense is that it’s likely more on the collective side, which doesn’t match your individual-centered research methods.

p.7. A map of your study site locations would be helpful.

p. 8 Please explain the Berns and Simpson 2009 reference a bit more. I would assume that people who work in extractive industries may spend lots of time in forests but not score high on conservation values.

p. 17 The last paragraph seems implicitly colonialist to me or at least vague about who is designing incentives and rules for resource governance. Also, “taking local values into account when designing incentives and rules for resource governance might lead to better outcomes” borders on banal. There’s plenty of justice-related arguments that already support this and decades of PES critiques that make this point repeatedly.

6. PLOS authors have the option to publish the peer review history of their article (what does this mean?). If published, this will include your full peer review and any attached files.

Reviewer #1: **Yes: **Sarah Klain

---

## [Author Response · Author response to Decision Letter 0]

12 Jun 2022

Dr. Klain,

We are appreciative of your thoughtful comments on our paper, “Conformity and tradition are more important than environmental values in constraining resource overharvest.” We found your critiques thought-provoking and we have done our best to incorporate your suggestions into our set of revisions. We believe the result is a stronger paper. In order to clarify our changes, we have outlined your comments below and described the changes we’ve made in response to each. 

Comment: I, however, see your research design as having a western, homo economicus approach to studying people outside of the “weird” cultures in which much of the theories underpinning political science and economics developed. I’m referencing Heinrich’s definition of “weird”: western, educated, industrialized, rich, democratic. The focus on individual values and differences at the individual level seem less relevant that a focus on identifying shared, collective values in your study sites. In the end, your data points to pro-social behavior driving decisions more than individually-oriented motivations driving decisions. In your set up and discussion of results, I think you ought to engage more with literatures that look beyond the individual, such as cultural ecosystem services, relational values and social valuation of the environment, e.g.:

Kenter, J. O., et al. (2016). Shared values and deliberative valuation: Future directions. Ecosystem Services, 21, 358–371. https://doi.org/10.1016/j.ecoser.2016.10.006

Also, I encourage you to peruse:

Levine, J., Chan, K. M. A. & Satterfield, T. (2015). From rational actor to efficient complexity manager: Exorcising the ghost of Homo economicus with a unified synthesis of cognition research. Ecological Economics, 114, 22–32. https://doi.org/10.1016/j.ecolecon.2015.03.010

Response: We appreciate these comments. It is quite true that we come from a political economy tradition based in rational choice theory (the “homo economicus” approach), though we hope that by expanding our inquiry to the role of values on resource governance we have stretched beyond those origins somewhat! And we have been careful to draw on a literature in social Psychology (the Schwartz theory of Values) which has been extensively tested outside of Western, Educated, Industrialized, Rich, and Democratic settings. 

We have included new language in the paper, in the Introduction (first paragraph), and the literature review (“Values, Environment, and Resource Management,” first, fourth, and fifth paragraph that engage with these questions, outlining these thoughts in somewhat greater detail and connecting our research to the literature on shared values as you suggest. 

The relationship between values held at the individual level (as measured by Schwartz) and the approach to shared value formation and elicitation (as outlined by Kenter and others) differ in some important (and perhaps paradoxical) ways. On the one hand, the Schwartz approach is methodologically individualistic in the sense that values are measured at the individual level, typically through survey instruments, and to the extent that values are measured at the group level by Schwartz and colleagues, those measured values are simply the aggregation of individual survey results. On the other hand, although the discursive approach to value elicitation and formation is definitively not based in the individual, much of the focus on the development and elicitation of values through discourse is related to decision-making process based in ecosystem services valuation. The notion of ecosystem services itself is controversial in some quarters because it too is perceived as too economistic, with scholars and practitioners essentially objecting to the idea that we can place a monetary value on ecosystems. We also have noted an additional criticism of this approach which is simply that ecosystem services valuation and similar processes are not widely used in many settings around the world, and even where discursive processes do not take place to elicit or develop shared values, policy-makers and resource users themselves hold values at the individual or group level; it seems important to us to examine whether these values have a measurable impact on resource management behaviors (hence our paper). 

Comment: Another interpretation of your results, which the Levine et al 2015 paper prompted me to recall, could be that people are efficiency maximizers. You evidence shows that identifying or not with pro-environmental values doesn’t explain behavior that well your study. Relying on tradition and conformity reduces cognitive burden and likely helps people maintain good relations in their small communities.

Response: This is a very interesting interpretation of our result, we would argue that this idea is complementary with our own interpretation, not necessarily a competing explanation; Collective action for conservation requires a shared understanding of the likely behaviors of other members of a particular community; if community members value tradition and conformity, they will likely have an easier time anticipating the actions of others because of a shared expectations around what is appropriate in a given situation. This might be both a way to efficiently manage complex situations (in a cognitive sense) and a way to enforce community norms (potentially by worsening interpersonal relationships in a predictable way when those norms are violated) in much the same way Ostrom and colleagues see institutions (rules, norms, and scripts) as ways to facilitate collective action. We have noted this helpful explanation in “Discussion” (paragraph 2).

Comment: p. 3 last line: Scripts needs some introduction.

Response: Thank you, we have included a definition as a note. 

Comment: p. 6 Explain more in the intro about why a game microcosm sheds light on real world behavior. I now see that you reflect on this on p. 15 but I’d like to see some of this in the study’s set-up. I remain skeptical that how people behave in a game when they know they are being studied is a good proxy for real world behavior. A brief review of literature on how game behavior provides insight on real world behavior related to environmental management could convince me otherwise. It seems like quite a reductionist approach to understanding real-world forestry management and choices. It’s also not clear to me if deciding the amount of trees to harvest in your study sites is a collective or individual choice. My sense is that it’s likely more on the collective side, which doesn’t match your individual-centered research methods.

Response: These are important questions which touch on broad debates in Political Science and Economics, among other areas. We would never argue that simulations, games or experiments like the one we present here are a replacement for other empirical methods, but these types of simulations allow us to control for many types of endogeneity that threaten causal inference in studies using other empirical techniques (which often demonstrate correlation while attempting to investigate causation). We have included a longer description of the ways games, simulations, and experiments have been used to study resource governance and related questions in “Methods” (paragraph 2). 

One interesting feature of our research design is that it allows us to examine both individual and collective choices and the link between the two; an important question we examine here is the extent to which values determine whether groups are able to make collective choices, i.e., whether individuals are more likely to stick to group agreements or to “cheat” and break agreements. In our simulation, harvesting decisions are made by individuals, but individuals within each group are free to communicate and therefore can (and often do) attempt to coordinate. However, like resource harvest in many remote settings, harvesting decisions are private and individual decisions are not immediately visible to other participants, so participants can “cheat” on those collective decisions (much like harvesting in real-world forest-dependent communities). Some groups are able to overcome these problems and act collectively while other are not—and of course our statistical analysis indicates that individuals with high levels of tradition and conformity are less likely to diverge from group agreements. We have inserted new language (paragraph 2 in “The forest management game” and paragraphs 9 and 10 in “Discussion”) to clarify these points. 

Comment: p.7. A map of your study site locations would be helpful.

Response: We have now added a map, currently Figure 1 on page 9. Thank you for this suggestion.

Comment: p. 8 Please explain the Berns and Simpson 2009 reference a bit more. I would assume that people who work in extractive industries may spend lots of time in forests but not score high on conservation values.

Response: This is a fair point and we suspect is related to the reason we found no relationship between forest visitation and any of our outcomes of interest. We initially spent little time discussing this citation (or the forest exposure/ecological knowledge literature more broadly) because we found no relationship between forest visitation and environmental values or behavior but we have added some additional language to elaborate on the Berns and Simpson reference. 

Comment: p. 17 The last paragraph seems implicitly colonialist to me or at least vague about who is designing incentives and rules for resource governance. Also, “taking local values into account when designing incentives and rules for resource governance might lead to better outcomes” borders on banal. There’s plenty of justice-related arguments that already support this and decades of PES critiques that make this point repeatedly.

Response: We didn’t intend this paragraph to sound colonialist, and though we agree that the point is (or should be) banal, it perhaps bears repeating since much of the NGO and IGO community continues to pursue policies which seek to impose Western/Northern models on developing communities. That said, we recognize Dr. Klain’s point and have substantially restructured the last paragraph to remove these sentences. 

We would again like to thank you for your helpful comments, and appreciate the opportunity to revise.

---

## [Decision Letter · Decision Letter 1]

19 Jul 2022

Conformity and tradition are more important than environmental values in constraining resource overharvest

PONE-D-21-34547R1

Dear Dr. Wright,

We’re pleased to inform you that your manuscript has been judged scientifically suitable for publication and will be formally accepted for publication once it meets all outstanding technical requirements.

Kind regards,

Hisham Zerriffi

Academic Editor

PLOS ONE

Additional Editor Comments (optional):

Reviewers' comments:

Reviewer's Responses to Questions

**Comments to the Author**

1. If the authors have adequately addressed your comments raised in a previous round of review and you feel that this manuscript is now acceptable for publication, you may indicate that here to bypass the “Comments to the Author” section, enter your conflict of interest statement in the “Confidential to Editor” section, and submit your "Accept" recommendation.

Reviewer #1: All comments have been addressed

2. Is the manuscript technically sound, and do the data support the conclusions?

Reviewer #1: Yes

3. Has the statistical analysis been performed appropriately and rigorously? 

Reviewer #1: Yes

4. Have the authors made all data underlying the findings in their manuscript fully available?

Reviewer #1: Yes

5. Is the manuscript presented in an intelligible fashion and written in standard English?

Reviewer #1: Yes

6. Review Comments to the Author

Reviewer #1: I appreciate your thoughtful incorporation of my previous feedback. This version is even more convincing and tailored to an interdisciplinary audience.

7. PLOS authors have the option to publish the peer review history of their article (what does this mean?). If published, this will include your full peer review and any attached files.

Reviewer #1: **Yes: **Sarah C. Klain

---

## [Editor Report · Acceptance letter]

9 Aug 2022

PONE-D-21-34547R1 

Conformity and tradition are more important than environmental values in constraining resource overharvest 

Dear Dr. Wright:

I'm pleased to inform you that your manuscript has been deemed suitable for publication in PLOS ONE. Congratulations! Your manuscript is now with our production department. 

Kind regards, 

on behalf of

Dr. Hisham Zerriffi 

Academic Editor

PLOS ONE